# Visualization of the Association of Dimeric Protein Complexes on Specific Enhancers in the Salivary Gland Nuclei of *Drosophila* Larva

**DOI:** 10.3390/cells13070613

**Published:** 2024-04-01

**Authors:** Solène Vanderperre, Samir Merabet

**Affiliations:** Institut de Génomique Fonctionnelle de Lyon (IGFL), UMR5242, Ecole Normale Supérieure de Lyon (ENSL), CNRS, Université de Lyon, 69007 Lyon, France; solene-vdp@hotmail.fr

**Keywords:** protein–protein interaction, BiFC, ANCHOR, ParB–INT, Hox

## Abstract

Transcription factors (TFs) regulate gene expression by recognizing specific target enhancers in the genome. The DNA-binding and regulatory activity of TFs depend on the presence of additional protein partners, leading to the formation of versatile and dynamic multimeric protein complexes. Visualizing these protein–protein interactions (PPIs) in the nucleus is key for decrypting the molecular cues underlying TF specificity in vivo. Over the last few years, Bimolecular Fluorescence Complementation (BiFC) has been developed in several model systems and applied in the analysis of different types of PPIs. In particular, BiFC has been applied when analyzing PPIs with hundreds of TFs in the nucleus of live *Drosophila* embryos. However, the visualization of PPIs at the level of specific target enhancers or genomic regions of interest awaits the advent of DNA-labelling methods that can be coupled with BiFC. Here, we present a novel experimental strategy that we have called BiFOR and that is based on the coupling of BiFC with the bacterial ANCHOR DNA-labelling system. We demonstrate that BiFOR enables the precise quantification of the enrichment of specific dimeric protein complexes on target enhancers in *Drosophila* salivary gland nuclei. Given its versatility and sensitivity, BiFOR could be applied more widely to other tissues during *Drosophila* development. Our work sets up the experimental basis for future applications of this strategy.

## 1. Introduction

Cell fate is determined by specific gene expression programs, which primarily occur at the level of transcription. This key regulatory process leads to the production of precise doses of mRNA molecules that are, subsequently, translated into proteins. Among the different cis-regulatory elements that control mRNA production are the so-called “enhancers”, which are DNA elements of relatively small size (less than 1 kilobase) that are located at various distances from the transcription start site [1]. Enhancers contain specific DNA sequences that are recognized by transcription factors (TFs). Importantly, the full activity of an enhancer relies on the combinatorial binding and activity of several TFs with their cofactors [2]. Therefore, understanding the molecular rules that govern enhancer activity requires the visualization of the protein–protein interactions (PPIs) that are specifically occurring on it, in the nucleus.

Several techniques have been developed to visualize the genomic regions of interest (ROIs). These techniques rely on the use of either fluorescent oligonucleotide probes for conducting fluorescent in situ hybridization (FISH), or DNA-binding proteins fused to fluorescent proteins (FPs). FISH provides a kilobase resolution, but requires stringent experimental procedures, such as DNA denaturation, that may affect PPIs ([3], see also the Section 3). Alternatively, different DNA-binding proteins have been fused to different FPs to target and label a genomic ROI in the nucleus. These strategies depend on multiplexing for getting enough FPs, bound to the target locus, for subsequent analysis using microscopy. For example, fluorescent dCas9 (catalytically dead Cas9) has been used to label repeated sequences in the genome [4,5,6]. Fluorescent dCas9 can also be used to label non-repetitive cis-regulatory elements by multiplexing guide RNAs (gRNAs) oligonucleotides [7]. However, these Cas9-based strategies have only been established in culture cells and await further development for applications in a multicellular organism. Other strategies exist to label genomic ROI in the nucleus. For example, the lactose operon repressor/operator (LacI/LacO) and the tetracycline operon repressor/operator (TetR/TetO) systems have been used to visualize chromosome translocations in live human cells [8]. Still, this system requires intensive engineering, since it is based on the insertion of a high number of LacO and TetO DNA sequences for exploiting the fluorescent signals emitted by the FP–LacI and FP–TetR fusion proteins. Another more promising approach is based on the Partition Protein B/Binding Sequence (ParB–INT or ANCHOR) system. In this system, multimerization of the fluorescent ParB proteins is achieved through PPIs on the *INT* sequence. First applied in yeast [9] and human cell culture [10,11], this system has also been developed to visualize enhancer–promoter interactions [12] or the DNA locus in *Drosophila* [13]. Moreover, recent analysis showed that the ANCHOR system did not locally perturb transcription in *Drosophila* (in contrast to the LacI/LacO system: [14]).

Here, we present a unique experimental approach coupling the ANCHOR and Bimolecular Fluorescence Complementation (BiFC) systems to visualize and measure the specific enrichment of dimeric protein complexes on a target DNA sequence in the nucleus of *Drosophila* larval salivary gland cells. BiFC relies on the properties of monomeric FPs to be reconstituted from two separate sub-fragments upon spatial proximity [15,16]. In particular, BiFC has been extensively used to reveal and analyze PPIs between various classes of TFs in live *Drosophila* embryos [17,18,19]. As a proof-of-concept, we used the well-characterized enhancer of the salivary gland selector gene *forkhead* (enhancer *fkh250*) as a regulatory model system. This enhancer is specifically regulated by the Hox protein Sex combs reduced (Scr) gene, in association with the Extradenticle (Exd) cofactor, in vivo [20]. This regulation occurs through the recognition of a unique Scr/Exd DNA-binding site. Importantly, Exd was shown to bind cooperatively with Scr, and not with any other Hox protein, on this enhancer, in vitro [20]. The specific regulation by Scr/Exd complexes was confirmed in vitro and in vivo with mutated enhancers in the Scr or Exd DNA-binding site [20]. Interestingly, the Scr/Exd DNA-binding site was also converted into a more consensus Hox/Exd binding site, allowing the binding of different Hox/Exd complexes in vitro and leading to ectopic expression in the embryo [20]. Altogether, the different variants of the *fkh250* enhancer constitute ideal tools for testing the specificity of the ANCHOR and BiFC systems. Our work demonstrates that BiFC and ANCHOR coupling (hereafter called “BiFOR”) is a powerful approach for the sensitive quantification of PPI enrichment on specific target enhancers in *Drosophila* salivary gland nuclei. 

## 2. Materials and Methods

### 2.1. Fly Lines

The salivary gland *sgs3*–*Gal4* driver was from the Bloomington stock center (line 6870 [21]). The *ParB1*–*mCherry* fly lines were provided by François Payre (CBI, Toulouse, France). 

The *UAS*–*VC*–*Ubx*, *UAS*–*-VC*–*Scr,* and *UAS*–*VN*–*Exd* fly lines were already established [22].

### 2.2. Generation of the INT1–fkh Enhancer Fly Lines

The different enhancers, *fkh250*, *fkh250_MUT_*, and *fkh250_CONS_*, were synthesized in ten copies (GenScript) and cloned between the EcoRI and XhoI sites, downstream of the three *INT1* sequences of the *pattB*–*INT1*–*hslacZ* vector (provided by F. Payre). All constructs were verified by sequencing, before sending for injection (BestGene Inc. Company, Chino Hills, CA, USA). The *INT1*–*fkh* constructs were all inserted in the same landing on the third chromosome, through the phiC31 integrase system [23].

### 2.3. Salivary Gland Preparation for Imaging

Dissected salivary glands were fixed at room temperature for 20 min in phosphate-buffered saline (PBS), containing 3.7% formaldehyde (formaldehyde methanol free, Thermo Fisher Scientific, Waltham, MA, USA), and washed 2 times for 30 min in 1× PBS. The salivary glands were either mounted in Vectashield with DAPI (Vector Labs) between a slide and a coverslip for directly visualizing the BiFC and ParB1–mCherry, or prepared for standard immunostaining, before mounting in order to reveal the VC–Ubx construct (using rabbit anti-GFP (A11122, Molecular Probes, Eugene, OR, USA, 1:500), recognizing the VC fragment [17], coupled to a secondary Alexa Fluor 488 (A11039, Molecular Probes)). 

### 2.4. FISH Probe Synthesis for the Forkhead Gene

The protocol for synthesizing DNA probes was taken from [24]. Briefly, a 12 kb long sequence centered on the *fkh* gene locus was chosen. Twelve pairs of 21 bp oligonucleotides were determined via the Primer3 website to cover the entire region (http://www.bioinformatics.nl/cgi-bin/primer3plus/primer3plus.cgi, accessed on 20 May 2020). Each fragment was then amplified by PCR from genomic DNA (Appendix A). Each fragment was approximately 1 kb long. The PCR products were then digested and labelled with the FISH Tag DNA orange kit (Alexa Fluor 555 dye, Thermo Fisher Scientific), following the manufacturer’s instructions.

### 2.5. Acquisition of Confocal Images for Lightning Deconvolution

The images were acquired using a confocal Leica SP8 microscope, with a 40× or 63× oil objective (HC PL APO 63×/1.40). The lightning option increases the confocal resolution thanks to the optimal acquisition parameters for the stack thickness, objective used, mounting medium, and laser wavelength. These parameters enable the maximum details to be extracted from each image voxel. The images acquired were oversampled (a large number of pixels) to enable processing. The image was scanned by a “LIGHTNING decision mask”, and various parameters were measured to determine the image quality of each pixel and adapt the processing according to the results. This differs from traditional methods, which apply reconstruction schemes with global rather than local efficiency. This mask removes much of the background noise and detects signals with a limited number of photons, making the image more resolute (theoretical resolution achieved: 120 nm). Therefore, the essential feature of the lightning results from the preservation of the photon number and the sum of all the intensities pre- and post-deconvolved images.

The acquisition parameters for each experimental condition were set (2656 × 2656 pixels; laser scan speed: 574 Hz; ×1.5 zoom). A 405 laser was used to capture an image of the DAPI. A 488 nm laser excites Venus for the BiFC signals, and a 552 nm laser excites mCherry to localize the *fkh250* enhancer (with a Leica HyD detector). See also Appendix A for the quantification pipeline of the fluorescent immunostaining and the BiFC on the *fkh* enhancers.

## 3. Results

### 3.1. Visualizing Hox/Exd Interactions by Conducting BiFC in Salivary Gland Nuclei

To validate that we could get exploitable BiFC signals in third instar larva (L3) salivary gland nuclei, we used previously published fly lines compatible with the UAS/Gal4 system. More precisely, Hox (Scr and Ultrabithorax, Ubx) and Exd constructs were fused to the C-terminal (UAS–VCScr and UAS–VCUbx) or N-terminal (UAS–VNExd) fragments of Venus [17,22]. The interaction between Hox and Exd fusion proteins should lead to the reconstitution of an excitable Venus fluorescent protein upon complementation between the VN and VC fragments (Figure 1A,B). The two UAS–VC–Hox constructs are inserted on the same landing site on the second chromosome, having comparable expression levels, as previously verified in the embryo [17,22]. The VC–Hox and VN–Exd fusion proteins were expressed in L3 salivary glands with the *sgs3*–*Gsal4* driver (see Section 2). The expression of VC–Hox constructs does not seem to affect polytene chromosomes, which look identical to polytene chromosomes of nuclei with no Hox-driven expression (Appendix A). As expected, immunofluorescent staining of VC–Scr and VC–Ubx with a GFP antibody recognizing the VC fragment (see Section 2) led to comparable fluorescence intensity in the salivary gland nuclei (Appendix A). However, the pattern was different, with VC–Ubx displaying more distinguishable bands on polytene chromosomes than VC–Scr. VC–Scr displayed a more diffusible and punctual pattern across the nucleus (Appendix A). It should be noticed that L3 salivary gland nuclei do not contain nuclear Exd given that Homothorax (Hth), which is required for the nuclear translocation of Exd, is transcriptionally repressed by Scr at early embryonic stages (by stage 11 [13]; see also https://flybase.org/reports/FBgn0001235, assessed on 22 February 2024) for RNA-seq data in L3 salivary glands). Thus, this observation suggests that Scr could bind less strongly on polytene chromosomes than Ubx in the Exd-free environment of L3 salivary gland nuclei. 

As expected, the two Hox proteins led to BiFC signals when co-expressed with Exd in the salivary gland nuclei (Figure 1A’,B’). However, the overall quantification of BiFC showed that the signals were three times higher on average with Ubx when compared to Scr (see Section 2 and Appendix A). This differential level of BiFC might be explained by the presence of endogenous Scr (although expressed at very low levels: https://flybase.org/reports/FBgn0003339, accessed on 22 February 2024), which could compete with VC–Scr for complex formation with VN–Exd. Not exclusively, the fusion topology may also affect the interaction between Scr and Exd, and, therefore, the BiFC signals. Most likely, the differential BiFC level could also reflect the ability of Ubx/Exd complexes to bind more widely on polytene chromosomes than Scr/Exd complexes. We hypothesize that it could be the case given that the patterns are different: Scr/Exd complexes are visible on a few but intense bands, while many more bands of variable intensities are observed with Ubx/Exd complexes (Figure 1A’,B’). Such differences in BiFC patterns also suggest that the fusion proteins are not expressed at saturation levels. 

In order to get better resolution signals, we applied an adaptative and automated deconvolution method to the confocal acquisitions (see Section 2). This manipulation revealed loci with a higher fluorescence intensity and with a pattern following the polytene chromosome organization for Ubx/Exd and to a lesser extent for Scr/Exd complexes (Figure 1A”,B”). Interestingly, the BiFC profile of Scr/Exd was also different from the binding profile of VC–Scr alone, with less widespread signals and a more defined pattern following the organization of polytene chromosomes. This different binding behavior suggests that Scr/Exd complexes are binding to more specific sites, with stronger affinity in the genome of salivary gland nuclei than the Scr monomers. This observation is in accordance with the role of Exd in forming a cooperative DNA-binding complex with Scr in vitro [20]. Moreover, BiFC signals are not systematically co-localized with DAPI (which depicts AT-rich and more condensed chromatin), underlining that Hox/Exd complexes are bound on different types of chromatin regions in salivary gland nuclei (enlargements in Figure 1A’,A”,B’,B”). Overall, these observations confirm that our fusion constructs and expression system are compatible for conducting BiFC in L3 salivary gland nuclei.

### 3.2. Experimental Conditions for DNA–FISH Are Detrimental for BiFC

We next wanted to assess whether we could quantify BiFC signals specifically on the *fkh* locus by conducting DNA–FISH conditions (Figure 2A,B). To this end, we designed fluorescent oligonucleotides spanning 12 kb upstream and downstream of the *fkh* gene (see Section 2) and performed FISH staining on BiFC-positive salivary gland nuclei. Although the FISH signals were clearly observable, we could not detect BiFC signals for Scr/Exd complexes (neither in the whole nucleus, nor on the *fkh* locus: Figure 2A’ and Appendix A). The BiFC signals were also extremely weak with Ubx/Exd complexes, making any quantification difficult. A more defined pattern of BiFC signals (Appendix A), especially after deconvolution (Figure 2B’), could be observed in a few nuclei. In these rare nuclei, the *fkh* FISH signal was systematically localized in a zone with low or no BiFC (Figure 2B’), which is in accordance with previous work showing that Ubx/Exd complexes can neither bind the *fkh250* enhancer in vitro, nor regulate its expression in vivo [20]. Altogether, these experiments reveal that the experimental conditions for FISH are detrimental for BiFC, underlining the need to use alternative approaches for conducting colocalization studies involving a genomic ROI. 

### 3.3. BiFOR Allows the Revelation of the Enrichment of Specific Dimeric Protein Complexes on a Target Enhancer in Salivary Gland Nuclei

Previous work established ANCHOR as a specific and sensitive method for labelling target ROIs in the genome, without affecting transcriptional regulation in vivo [13,14]. 

We, therefore, decided to couple *INT1* sequences to different versions of the *fkh250* enhancer: wild type, mutant (with mutations abolishing the DNA binding of Scr and Exd: *fkh250_MUT_*), and consensus (with mutations transforming the specific Scr/Exd into a generic Hox/Exd binding site and allowing recognition by different Hox/Exd complexes: *fkh_CONS_*). These mutations were based on previous work on *fkh250* regulation by Hox/Exd complexes [20]. The *INT1*–*fkh250* fusion constructs were all inserted at the same landing site in the genome ([23] and Section 2). ParB1–mCherry fusion proteins were expressed in the salivary glands with the UAS/Gal4 system to label the different *INT1–fkh250* fusion constructs (Section 2). All *INT1–fkh250* fusion constructs were specifically recognized by ParB1–mCherry, leading to a strong red fluorescent signal in the nucleus. This signal can be of a variable size and shape depending on the orientation of the nucleus, and was also usually less intense in the case of the *INT1–fkh250_MUT_* enhancer construct. 

We first performed BiFC by expressing VC–Scr and VN–Exd in ParB1–mCherry/*INT1*–*fkh250*-positive salivary gland nuclei (Figure 3A). In contrast to the previous observation under the DNA–FISH conditions, the VC–Scr/VN–Exd BiFC signals were not affected. Quantification (see also Section 2) showed that the BiFC signals were significantly enriched with ParB1–mCherry, demonstrating preferential localization on the *fkh250* enhancer when compared to other genomic loci in the salivary gland nuclei (Figure 3A’,A” and Appendix A).

This observation is in accordance with the specific recognition and regulation of *fkh250* by Scr/Exd in vitro and in vivo [20]. As a first negative control experiment, we repeated the analysis by expressing VC–Ubx in combination with VN–Exd. As expected, the VC–Ubx/VN–Exd signals were stronger globally than the ones obtained with VC–Scr/VN–Exd in salivary gland nuclei. Still, quantification showed a significant diminution of enrichment on the *fkh250* enhancer (Figure 3B’,B”). We performed additional negative control experiments by expressing VC–Ubx or VC–Scr alone, which do not or only poorly bind on *fkh250* in vitro, respectively [20]. The expression of VC–Scr or VC–Ubx led to more, albeit less defined, signals along the polytene chromosomes than the corresponding BiFC complexes with Exd (Figure 3C’,D’). Still, quantification showed the absence of a significant enrichment on the *INT1*–*fkh250* enhancers when compared to Scr/Exd complexes (Figure 3C”,D”). This observation underlines that neither Scr, nor Ubx, alone or in complexes with Exd could preferentially bind the *fkh250* enhancer in salivary gland nuclei. This result is in accordance with previous observations in vitro [20]. Together with the specific enrichment of VC–Scr/VN–Exd BiFC, our results indicate that BiFOR can recapitulate the specific recognition of target enhancers through a dimeric protein complex in salivary gland nuclei. 

### 3.4. BiFOR Allows the Revelation of Dimeric Complex Enrichment on Specific Target Enhancers in Salivary Gland Nuclei

To further confirm the specificity of BiFOR, we generated two additional variants of the *fkh250* enhancer fused to the *INT1* sequences: *fkh250_MUT_* and *fkh250_CONS_*. As previously mentioned, mutations in the *flh250_MUT_* enhancer abolish Hox/Exd binding in vitro [20]. Not surprisingly, the analysis in salivary gland nuclei showed the absence of any significant enrichment of BiFC resulting from Scr/Exd complex assembly at the level of the *fkh250_MUT_*-containing genomic locus (Figure 4A–A”). In contrast, the Scr/Exd and Ubx/Exd BiFC signals were similarly and significantly enriched on *fkh250_CONS_* (Figure 4B–B”,C–C”), confirming the previous in vitro observations related to the gained ability of this modified enhancer to be recognized by different Hox/Exd complexes [20]. 

In conclusion, both the use of different Hox/Exd complexes and variants of the *fkh250* enhancer confirmed the specificity and sensitivity of BiFOR for visualizing and quantifying PPI enrichment on a target enhancer of interest in salivary gland nuclei. 

## 4. Discussion

### 4.1. Advantages of ANCHOR over Other DNA-Labelling Systems for BiFC Coupling

Several techniques exist to visualize genomic ROIs in the nucleus. Here, we first tried to perform FISH for analyzing the BiFC on the genomic locus of *fkh*. We found that the experimental conditions of FISH were detrimental for BiFC, forbidding any quantification measurements and underlining the need to use alternative and softer DNA-labelling approaches. 

Several alternative methods have been developed to label genomic ROIs, the main challenge being to have enough specific fluorescent signals for proper quantification in the nucleus. Among them are Crispr/Cas9-based approaches, which are long and labor intensive for the cloning and expression of multiple copies of specific gRNAs. These issues might explain why Crispr/Cas9-based DNA-labeling approaches have not yet been applied in a multicellular organism [4,5,6,7]. The LacO/LacI system has been used in *Drosophila*, but it also requires a hundred copies of the LacO cassette to exploit the fluorescent signals emitted by the DNA-bound LacI–FP fusion protein [8]. In addition, the binding of LacI has been shown to interfere with gene transcription and is, therefore, not the most appropriate option for quantifying DNA-bound TFs in the genomic vicinity [14]. In contrast, the ANCHOR system is more neutral in regard to gene transcription [14]. Moreover, the property of ParB proteins to oligomerize on ParS binding sites through protein–protein interactions allows the observation of the ParB–FP without multiplying the number of ParS cassettes (usually three copies are used [9]). The existence of different ParB proteins recognizing different ParS sequences (ParB1/ParS1 and ParB2/ParS2, for instance) also allows the visualization of two different loci simultaneously, when each ParB protein is fused to a different FP [14]. Thus, the ANCHOR system appears to be a quite straightforward and neutral DNA-labelling system for visualizing genomic ROIs in the nucleus. Here, we demonstrated that BiFC could be coupled with ANCHOR to visualize PPIs on a specific target enhancer in whole salivary gland nuclei. Moreover, since BiFC stabilizes the dimeric complex upon complementation (with the formation of covalent bonds between the two sub-fragments of the FP), it allows the analysis not only of strong and stable PPIs, but also weak and transient PPIs in a ROI. 

### 4.2. Using BiFC to Visualize Dimeric Complexes in Whole-Mounted Salivary Gland Nuclei

Given the weak affinity between complementary split fragments, BiFC could potentially reveal artificial interactions when two fusion proteins are too strongly expressed. It is, therefore, critical to know the expression level and use negative controls, when possible. The best negative controls consist in using mutated versions that affect the interaction. Alternative controls like those expressing one of the candidate partners as a cold competitor against BiFC could also have been used. 

Here, we used the *Sgs3*–*Gal4* driver, which is specifically expressed at the mid-third instar transition in salivary glands [21]. This driver is more specific than the other salivary gland drivers that are expressed at earlier developmental stages and in other tissues (like *fkh*–*Gal4* [25]). It is, therefore, the most appropriate option for conducting BiFC in L3 salivary glands. Hox fusion proteins were inserted at the same genomic landing site, having comparable expression levels. Negative BiFC controls, with Hox and Exd fusion proteins that cannot interact together under Hox-like strong expression levels in the embryo, have already been used elsewhere [22]. In addition, despite the strong expression of *Sgs3*–*Gal4*, the distinct binding patterns on polytene chromosomes with VC–Scr and VC–Ubx, either alone or in combination with VN–Exd, underlines that our expression system was not at the saturation level in salivary gland nuclei. This conclusion is further reinforced by the observation of specific enrichments on different versions of the *fkh250* enhancer, despite the size of the target locus (see below). Along the same lines, BiFC has previously been used for visualizing the genome-wide binding behavior of a specific dimeric complex on polytene chromosome spreads [26]. BiFOR could potentially have been performed on squashed polytene chromosomes, making the confocal acquisitions of better resolution and quality. However, the purpose of our work was to develop an approach that allows the visualization of dimeric complexes on a specific genomic ROI at a high resolution, when considering the three dimensional space of the whole nucleus. In this context, it is important to maintain the native context of chromosomes, allowing the preservation of important molecular features like intra- or inter-chromosome contacts, for example. 

Finally, our results were obtained on fixed salivary glands. Here, the fixation was dictated by the long acquisition time required for lightning deconvolution (between 40 min and one hour long on average for a two-color acquisition). Without such a need for a high resolution, BiFOR could be used on live salivary glands using conventional confocal microscopy (with a few minutes long acquisition time). Existing post-acquisition deconvolution algorithms could have been applied to improve the resolution (although to a lesser extent than lightning or other deconvolution methods that require specific acquisition parameters). 

### 4.3. Futures Perspectives on BiFOR 

Our system was based on transgenic constructs, with ten copies of wild-type or modified *fkh* enhancers to ensure the visualization of BiFC signal enrichment in salivary gland nuclei. Salivary gland nuclei are characterized by multi-replicated polytene chromosomes, which also favors the visualization of TF binding events. Our analysis revealed clear and strong signals for both ParB1–mCherry and BiFC, which cover approximately between 40 and 50 kb. Still, the system was sensitive enough to reveal significant differences between the different protein and enhancer combinations, and to recapitulate previous observations using purified proteins on single DNA-binding sites in vitro [20].

Compared to large-scale approaches, such as ChIP-Seq or RNA-Seq, BiFOR offers complementary information for validating specific regulation and binding on the candidate target enhancer. For example, in silico analyses of cis-regulatory sequences of a TF target gene often provides a list of associated candidate TFs that could also bind and participate in the transcriptional regulation. BiFOR opens up the possibility to validate whether a candidate TF could interact and form enriched dimeric complexes with the regulatory TF on the candidate target enhancer identified from ChIP-Seq and RNA-Seq analyses. Hundreds of TFs are available for conducting BiFC analyses with the UAS/Gal4 expression system [19]. The *INT1*-enhancer construct could be inserted into the genome, as shown in this work. Given the intensity of ParB1–mCherry and BiFC signals observed with the *fkh250* enhancers, we anticipate that one copy of *INT1* will be enough and could even be inserted in close proximity to the endogenous candidate enhancer to directly visualize the ROI in salivary gland nuclei. 

The intensity of fluorescent signals also suggests that BiFOR could be exploited in diploid tissues, although it remains to be demonstrated. Along these lines, the ANCHOR system has previously been shown to efficiently label a genomic ROI in *Drosophila* imaginal discs [14]. The exploitation of BiFC might depend on the number of DNA-binding sites in the target enhancer. For example, *fkh250* contains only one Hox/Exd binding site, and a minimum number of copies is probably required to get exploitable BiFC signals in diploid nuclei under a conventional confocal microscope resolution. An alternative strategy to further increase the intensity and/or resolution of BiFC signals in this context could be the use of a BiFC-specific nanobody [27]. Indeed, the reconstitution of the FP upon complementation provides only 20% of the brightness of the original FP used for BiFC. This issue could be circumvented by using a BiFC-specific nanobody fused to a bright FP (such as mNeonGreen [28]) or other fluorescence-emitting proteins (such as HaloTag [29]). These imaging tools are compatible with STED microscopy and could, therefore, also increase the resolution scale of PPI analysis on the target enhancer. The same rationale could apply to the ANCHOR labelling system. Here, we used ParB1–mCherry fusion proteins, but other ParB proteins that fuse to brighter fluorescent proteins or systems are available [14]. 

## 5. Conclusions

Our work established the coupling of BiFC and ANCHOR for visualizing dimeric protein complexes on a genomic ROI in salivary gland nuclei. This work sets the experimental foundation for future technological developments aimed at visualizing the association of dimeric protein complexes on specific endogenous loci in diploid tissues. These future developments will offer new perspectives for understanding transcriptional specificity in vivo. The versatility of BiFC and ANCHOR tools makes the BiFOR approach suitable for *Drosophila* and other model organisms.

## Figures and Tables

**Figure 1 cells-13-00613-f001:**
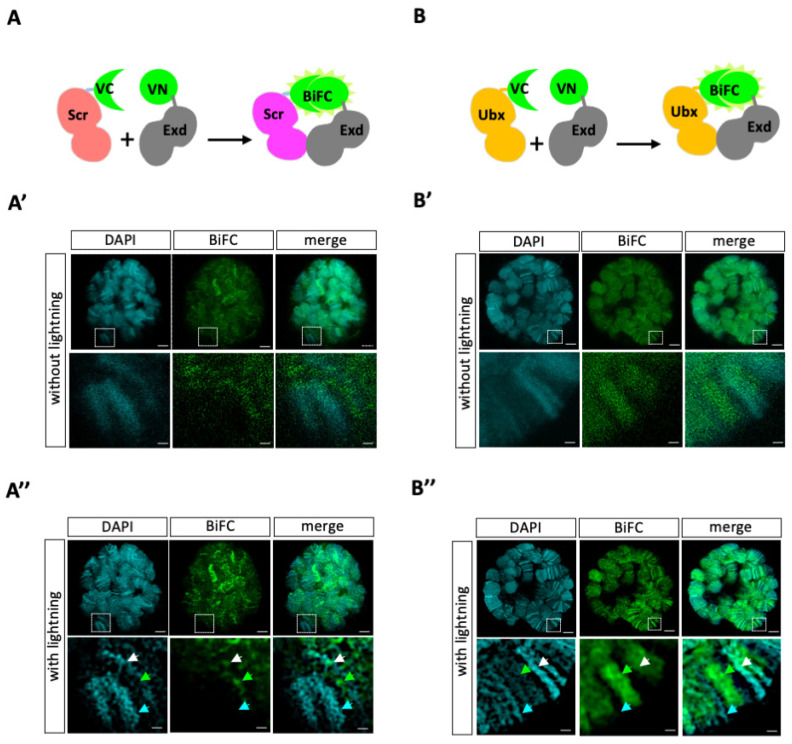
Visualization of BiFC signals in L3 salivary gland nuclei. (**A**,**B**). Principle of the BiFC obtained with the VC–Scr and VN–Exd (A) or VC–Ubx and VN–Exd (B) constructs. (**A’**,**B’**). Illustrative L3 salivary gland nucleus with the BiFC (green) between Scr and Exd (**A’**) or Ubx and Exd (**B’**), with DAPI staining of the polytene chromosomes (cyan). Enlargement of a particular zone is shown to highlight both overlapping and non-overlapping staining between BiFC and DAPI. Images involve confocal acquisition without lightning deconvolution. (**A”**,**B”**). The same illustrative salivary gland nuclei as in (**A’**,**B’**) but with applied lightning acquisition parameters and deconvolution to improve the resolution and signal-to-noise ratio (see also Section 2). Cyan, green, and white arrows highlight regions with DAPI only, BiFC only, or DAPI and BiFC merged signals, respectively. VN: N-terminal fragments of Venus (1–172). VC: C-terminal fragments of Venus (155–235). Scale bars = 5 μm (upper panels) or 1 μm (enlargements).

**Figure 2 cells-13-00613-f002:**
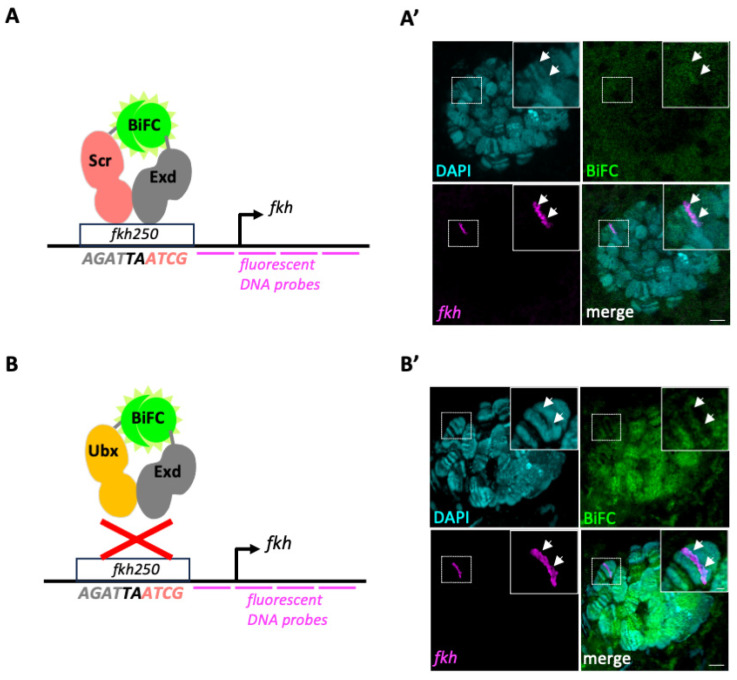
Experimental conditions for DNA–FISH are detrimental to BiFC. (**A**,**B**). Principle of the coupling between BiFC resulting from VC–Scr/VN–Exd (**A**) or VC–Ubx/VN–Exd (**B**) complex assembly and FISH staining of the *forkhead* (*fkh*) genomic locus. The nucleotide sequence of the DNA-binding site that is specifically recognized by Scr/Exd complexes in the *fkh250* enhancer is indicated. Gray and light red nucleotides are bound by the Exd or the Hox protein, respectively. The two central nucleotides (black) are recognized by the two proteins (based on [20]). The red cross illustrates the inability of Ubx/Exd complexes to bind on the *fkh250* enhancer in vitro (based on [20]) (**A’**,**B’**). Illustrative L3 salivary gland nucleus with BiFC (green) resulting from VC–Scr/VN–Exd (**A’**) or VC–Ubx/VN–Exd (**B’**) complex assembly, FISH of the *fkh* locus (magenta), and DAPI staining of polytene chromosomes (cyan). No exploitable BiFC signals are obtained under FISH experimental conditions for VC–Scr/VN–Exd complexes. Very few nuclei showed a weak BiFC pattern with VC–Ubx/VN–Exd. The enlargement in (**B’**) highlights the non-overlapping pattern between the BiFC and FISH signals (white arrows). Images involve confocal acquisition with lightning deconvolution (see also Appendix A). Scale bars = 5 μm (upper panels) or 1 μm (enlargements).

**Figure 3 cells-13-00613-f003:**
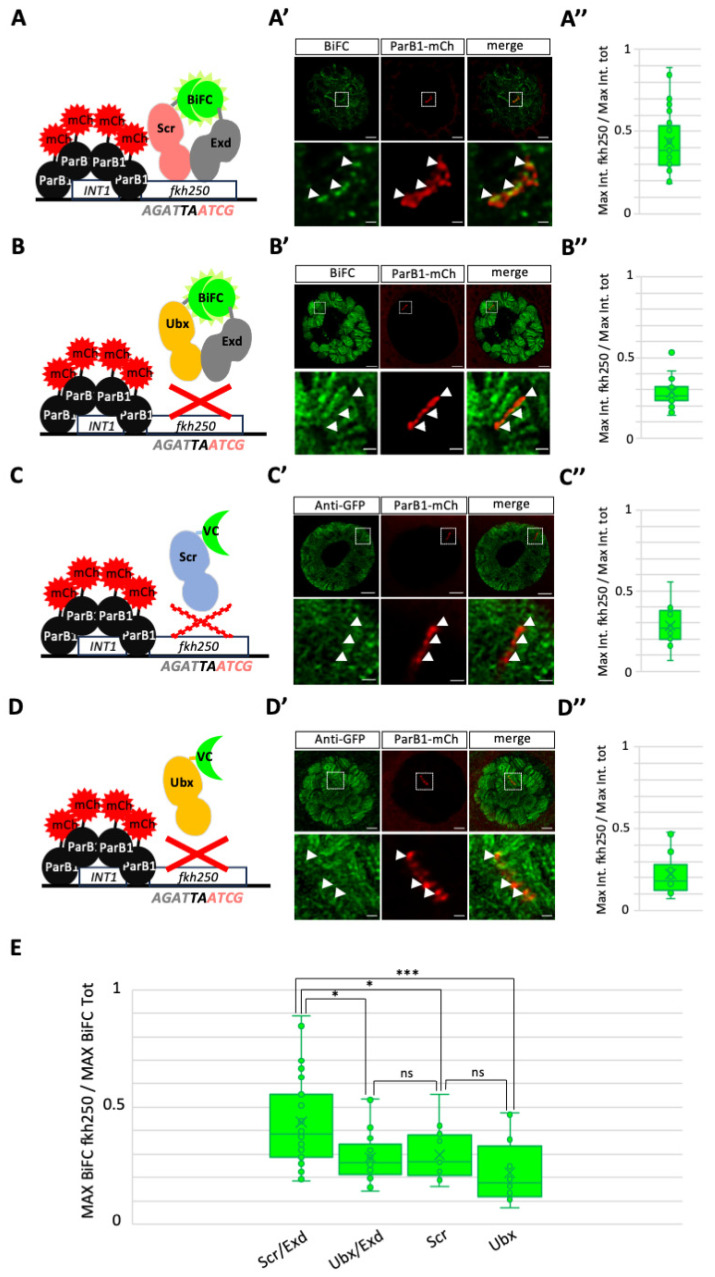
Assessing the specificity of BiFOR with the wild-type *fkh250* enhancer in salivary gland nuclei. (**A**–**D**). Schematic representation of the different conditions. The ParB1–mCherry (mCh) fusion proteins oligomerize on the target *INT1* cassettes that are inserted close to the *fkh250* enhancer (see Section 2). Only Scr/Exd complexes were shown to cooperatively bind on *fkh250* in vitro [20]. No cooperative binding was observed for Ubx/Exd complexes (but Exd improves monomer binding of Ubx in vitro [20]). Extremely weak (illustrated with a dotted red cross) or no (illustrated with a red cross) binding in vitro was observed for Scr and Ubx as a monomer, respectively [20]. (**A’**–**D’**). Illustrative confocal acquisition with lightning deconvolution of salivary gland nuclei expressing the different constructs, as indicated. Protein enrichment (green) was analyzed at the level of the inserted *fkh250* enhancers (red) and compared to the overall maximum green fluorescence intensity in the nucleus (see Section 2). Enlargements are shown at the level of the ParB1–mCherry signal to better highlight the presence or absence of colocalization (white arrowheads) with BiFC (**A’**,**B’**) or VC–Scr and VC–Ubx (**C’**,**D’**), revealed with an anti-GFP antibody recognizing the VC fragment (see Section 2). (**A”**–**D”**). Statistical quantification of BiFC (**A”**,**B”**) or immunostaining (**C”**,**D”**) at the level of the ParB1–mCherry signal. Values were obtained from the maximum fluorescence intensity measured on the *fkh250* enhancer, divided by the maximum fluorescence intensity measured in the whole nucleus. (**E**). Statistical comparative analyses of the different conditions on *fkh250*, as indicated. The *p* values are obtained from three independent biological replicates. Notes: * *p* values ≤ 0.01, *** *p* value ≤ 0.0001, ns: not significant. Scale bars = 5 μm (upper panels) or 1 μm (enlargements). See also Appendix A.

**Figure 4 cells-13-00613-f004:**
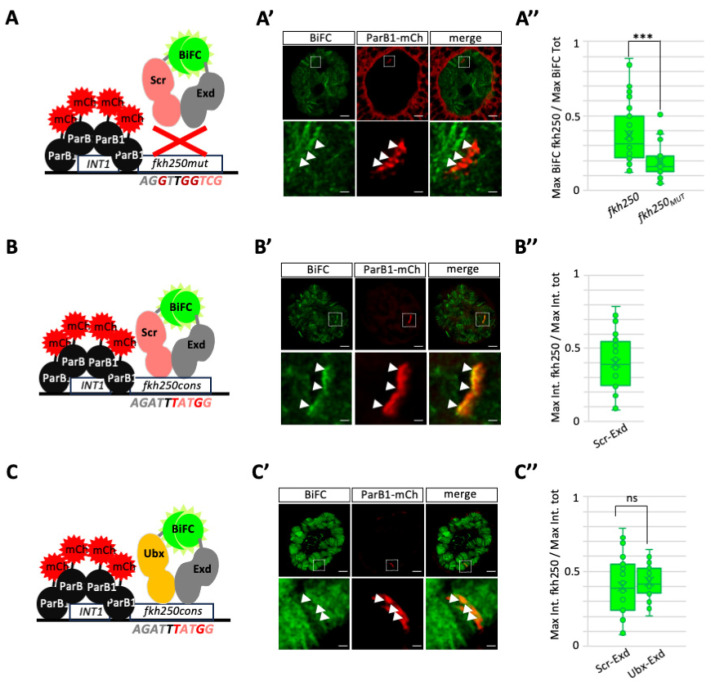
Assessing the specificity of BiFOR on two different variants of the *fkh250* enhancer in salivary gland nuclei. (**A**–**C**). Schematic representation of the *fkh250_MUT_* (**A**) and *fkh250_CONS_* (**B**,**C**) enhancer variants. Mutated residues in the Hox and/or Exd DNA-binding site are highlighted in bold red. Their negative (red cross) or positive effect on Hox/Exd binding is based on previous in vitro and in vivo studies [20]. (**A’**–**C’**). Illustrative confocal acquisition with lightning deconvolution of salivary gland nuclei expressing the different constructs, as indicated. BiFC enrichment (green) was analyzed at the level of the *fkh250_MUT_* (**A’**) or *fkh250_CONS_* (**B’**,**C’**) enhancers (labelled with ParB1–mCherry, red) and compared to the overall maximum green fluorescence intensity in the nucleus (see Section 2). Enlargements are shown at the level of the ParB1–mCherry signal to better highlight the absence (**A’**, white arrows) or presence (**B’**,**C’**, white arrows) of colocalized green and red signals. (**A”**–**C”**). Statistical quantification of BiFC at the level of the ParB1–mCherry signal. Values were obtained from the maximum fluorescence intensity measured on the mutant (**A”**) or consensus (**B”**,**C”**) *fkh250* enhancer, divided by the maximum fluorescence intensity measured in the whole nucleus. Quantification is obtained from three independent biological replicates. Notes: *** *p* value ≤ 0.0001, ns: not significant. Scale bars = 5 μm (upper panels) or 1 μm (enlargements). See also Appendix A.

## Data Availability

Dataset available on request from the authors.

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
