# Peer review of "Visualization of the Association of Dimeric Protein Complexes on Specific Enhancers in the Salivary Gland Nuclei of Drosophila Larva"

_cells, 2024, doi:10.3390/cells13070613_

Round 1
Reviewer 1 Report
Comments and Suggestions for Authors
Summary
Tractable methods to probe protein complexes in vivo still remain relatively elusive in biology. In this manuscript, Vanderperre and Merabet demonstrate that Bimolecular Fluorescence Complementation (BiFC) combined with the ANCHOR DNA-labelling method can be used to visualize heterodimeric Hox proteins assembling on DNA in vivo. Using the drosophila salivary gland as a model, the manuscript shows the limitations of smFISH with BiFC and how ANCHOR can instead be used to fluorescently mark a particular locus of interest. Using the Ubx, Scr, and Exd Hox transcription factors, they show how BiFC tagged protein versions can be visualized in the salivary gland nuclei, and how the Scr/Exd pairing specifically assembles with the ANCHOR system, as designed. Mutations to the transcription binding site at the reporter prevents Scr/Exd colocalization.
The manuscript was well written, and the presented results were moderately clear. Some figures should be revised to clarify the experimental set up, but this criticism is relatively minor. The method itself is a development, albeit a modest one without new biological insight. While the manuscript conclusions were reasonable, the value of the method for future applications is unclear without more controls. Specifically, it remained unclear whether Scr alone is capable of localizing to the reporter gene without the help of ExD. With further clarity, the work will be of broad interest to cell biologists and those studying protein-protein interactions.
MAJOR
1. The method and manuscript will be greatly strengthened with more controls For example: a) a negative fluorescent control of Scr/Exd without N- or C- BiFC tags, and b) Scr/VN-Exd and/or VC-Scr/Exd. A concern is that we are just visualizing Scr-dependent localization versus Scr/Exd localization. (see #2 comment).
2. The manuscript makes its point that the BiFC fluorescence visualized at the reporter loci of interest with the proper Scr/Exd pairing, but not with a comparable Ubx/Exd pairing. The BiFC fluorescence also indicates that the two proteins are tied together in a complex. The results do not clarify whether the Scr/Exd proteins are cooperatively binding DNA, or if just the Scr component is binding and bringing along the other for a ride. The manuscript and method will be greatly strengthened with controls measuring the sole dependence on Scr for fluorescence recruitment to the reporter loci.
MINOR
1. The red X in Fig 2-4 should be defined.
2. What are the protein levels of each transcription factor component? The result may help explain the BiFC fluorescent differences between Scr and Ubx.
3. The fluorescence signal from BiFC versus antibody staining is not clear in Figure 3.
Author Response
MAJOR
- The method and manuscript will be greatly strengthened with more controls For example: a) a negative fluorescent control of Scr/Exd without N- or C- BiFC tags, and b) Scr/VN-Exd and/or VC-Scr/Exd. A concern is that we are just visualizing Scr-dependent localization versus Scr/Exd localization. (see #2 comment).
- The manuscript makes its point that the BiFC fluorescence visualized at the reporter loci of interest with the proper Scr/Exd pairing, but not with a comparable Ubx/Exd pairing. The BiFC fluorescence also indicates that the two proteins are tied together in a complex. The results do not clarify whether the Scr/Exd proteins are cooperatively binding DNA, or if just the Scr component is binding and bringing along the other for a ride. The manuscript and method will be greatly strengthened with controls measuring the sole dependence on Scr for fluorescence recruitment to the reporter loci.
We agree with the reviewer and now present the binding profile of VC-Scr alone in the revised version (new Fig. S1, S5 and new Fig. 3). These novel experiments showed clear distinct binding behaviors between Scr and Ubx alone (new paragraph: lanes 167-178), and between Scr and Scr/Exd complexes (new paragraph: lanes 218-224). The cooperative DNA-binding activity of Scr/Exd complexes in vitro was also more clearly stated in the Introduction section (lanes 73-75).
MINOR
- The red X in Fig 2-4 should be defined.
We have now defined more clearly the red cross in the legend of Figures 2-4.
- What are the protein levels of each transcription factor component? The result may help explain the BiFC fluorescent differences between Scr and Ubx.
The experiments with VC-Scr or VC-Ubx alone now enable to conclude on this aspect. We have rewritten the related part in the results section (lanes 167-173). We confirmed that the constructs are expressed at similar level (new Fig S1) but behave differently on polytene chromosomes (with a stronger binding of Ubx monomers).
- The fluorescence signal from BiFC versus antibody staining is not clear in Figure 3.
We have changed the label of the antibody staining as “anti-GFP” to make it clearer.
Reviewer 2 Report
Comments and Suggestions for Authors
The manuscript is devoted to an interesting technique designed to solve a problem of visualization of dimeric protein complex association on sites of interest in salivary gland nuclei of the Drosophila larva. It should be emphasized that the article is methodological, its goal is to show that the described system is fundamentally suitable for use. All experiments illustrate interactions previously shown by other methods.
The article is written clearly and logically.
At the same time, I believe that some issues need to be clarified before the article can be published.
The method is based on very strong overexpression of proteins (the sgs3 driver provides a very high level of expression). The question arises as to how much the presence of a signal reflects any native situation. Generally the excess of each protein leads to an increase in the number of potential binding sites. Is the problem stated by the authors: " to precisely quantify the enrichment of specific dimeric protein complexes on target enhancers in Drosophila salivary gland nuclei " really solvable in such a system?
In the photographs in the manuscript under review the number of sites is comparable to the number of polytene chromosome bands medium and large in size, which significantly exceeds the number of sites of the corresponding transcription factors normally. Additionally the work is done on uncrushed whole nuclei, the chromosomes lie quite compactly, and the level of resolution by cytological methods is about 50 kb. Therefore, the presence of a signal does not seem very informative. Only the complete absence of a signal indicates that protein pairs, even in excess, do not land on the region of interest.
In this regard, I would like the authors to discuss the limitations of the method associated with protein overexpression and resolution.
The problem of artifactual protein interactions arising as a result of overexpression is discussed, for example, in the article: Sampaio et al., Investigation of interactions between TLR2, MyD88 and TIRAP by bioluminescence resonance energy transfer is hampered by artefacts of protein overexpression. PLoS One. 2018 Aug 23;13(8):e0202408. doi: 10.1371/journal.pone.0202408.
Since the article is methodological, I would like the authors to discuss what fundamentally new information this method will provide in comparison with genomic analysis, for example, Chip Seq.
I would like the authors to explain why they focus on whole cells, where complex cytological approaches like ANCOR are required to detect a site of interest, and did not try to work with squashed salivary glands. At first glance, squashed ones should be better, since there 1) the site can be seen from the pattern of polytene chromosome bands without additional tricks 2) the resolution is significantly higher, and on well-squashed chromosomes it is possible to distinguish two marks at a distance of about 5 kb 3) for FISH you can use 500 bp probes instead of 12 kb, as in the manuscript 4) you can also see the GFP-like fluorwacwnt proteins with adequate fixation methods (discussed in Johansen et al Polytene chromosome squash methods for studying transcription and epigenetic chromatin modification in Drosophila using antibodies. Methods. 2009 Aug;48(4):387-97.)
BiFC and ANCHOR DNA-labelling system allow in vivo visualization. Can the authors expect the development of an approach for in vivo microscopy?
The authors write “It should be noticed that L3 salivary gland nuclei do not contain endogenous Exd given that its nuclear transporter Homothorax (Hth) is repressed by Scr “
It is not entirely clear whether such an effect of Src on the nuclear transport of Exd will be manifested in an ectopic expression system. Could it be this effect that explains the differences in the amount of signal between Scr and Ubx (if Scr suppresses Exd transport, but Ubx does not)?
Figure 1 shows the result, which even after deconvolution looks quite fuzzy. Many questions arise about the proof of the specificity of this signal. It would be useful to show what the chromosomes look like when each construct is overexpressed separately.
It is noteworthy that in Figure 3 there is a much clearer BiFOR signal for Ubx than in Figure 1
It would be useful to show the DAPI staining in Figure 3.
In all figures, it would be helpful to add labels (such as arrows) to indicate where the signal is located on all color channels. The colors used when superimposing do not always make it possible to understand how the signals relate in space. For example, when superimposing green on blue in Figure 2, it is not clear where there is colocalization and where there is not. Arrows would help understand the correspondence of signals even without overlap.
The authors write “ In order to get a better resolution of signals, we applied an adaptative and automated deconvolution method on confocal acquisitions (see materials and methods).” The materials and methods do not contain the word deconvolution; it is not entirely clear that what is described in the materials and methods corresponds to the adaptive and automated deconvolution method
Comments on the Quality of English Language
I am not an expert in English, but it seemed to me that the text is written in good English. I noticed only a few typos, for example, there is a period in one of the headings (2.3. Salivary gland preparation for imaging. ). ФC-31 is usually written differently. But in general I have no complaints about the English language.
Author Response
At the same time, I believe that some issues need to be clarified before the article can be published.
The method is based on very strong overexpression of proteins (the sgs3 driver provides a very high level of expression). The question arises as to how much the presence of a signal reflects any native situation. Generally the excess of each protein leads to an increase in the number of potential binding sites. Is the problem stated by the authors: " to precisely quantify the enrichment of specific dimeric protein complexes on target enhancers in Drosophila salivary gland nuclei " really solvable in such a system?
In the photographs in the manuscript under review the number of sites is comparable to the number of polytene chromosome bands medium and large in size, which significantly exceeds the number of sites of the corresponding transcription factors normally. Additionally the work is done on uncrushed whole nuclei, the chromosomes lie quite compactly, and the level of resolution by cytological methods is about 50 kb. Therefore, the presence of a signal does not seem very informative. Only the complete absence of a signal indicates that protein pairs, even in excess, do not land on the region of interest.
In this regard, I would like the authors to discuss the limitations of the method associated with protein overexpression and resolution.
The problem of artifactual protein interactions arising as a result of overexpression is discussed, for example, in the article: Sampaio et al., Investigation of interactions between TLR2, MyD88 and TIRAP by bioluminescence resonance energy transfer is hampered by artefacts of protein overexpression. PLoS One. 2018 Aug 23;13(8):e0202408. doi: 10.1371/journal.pone.0202408.
These comments relate to the level of expression of the fusion proteins with regard to their protein-protein interaction properties and DNA-binding behavior on polytene chromosomes. We agree with the reviewer that expression levels are critical and we tried to clarify better our expression system with this regard. First, we emphasized the different DNA-binding behaviors of monomers Hox versus Hox/Exd dimers to highlight that our expression system is not at saturation (lanes 208-213 in the Results section and lanes 714-717 in the Discussion section). Second, we provided more information on the sgs3-Gal3 driver (which is a late expressing driver in L3 salivary glands: lanes 707-709 in the Discussion section). Third, we cited previous work showing that negative controls validated the fusion proteins used in this study for doing BiFC (with stronger Hox-Gal4 drivers; lanes 712-714 in the Discussion section).
Since the article is methodological, I would like the authors to discuss what fundamentally new information this method will provide in comparison with genomic analysis, for example, Chip Seq.
We thank the reviewer for this suggestion and wrote a dedicated part in the Discussion section (lanes 745-757).
I would like the authors to explain why they focus on whole cells, where complex cytological approaches like ANCOR are required to detect a site of interest, and did not try to work with squashed salivary glands. At first glance, squashed ones should be better, since there 1) the site can be seen from the pattern of polytene chromosome bands without additional tricks 2) the resolution is significantly higher, and on well-squashed chromosomes it is possible to distinguish two marks at a distance of about 5 kb 3) for FISH you can use 500 bp probes instead of 12 kb, as in the manuscript 4) you can also see the GFP-like fluorwacwnt proteins with adequate fixation methods (discussed in Johansen et al Polytene chromosome squash methods for studying transcription and epigenetic chromatin modification in Drosophila using antibodies. Methods. 2009 Aug;48(4):387-97.)
We made a specific point on this aspect in the new paragraph of the Discussion section (lanes 721-727).
BiFC and ANCHOR DNA-labelling system allow in vivo visualization. Can the authors expect the development of an approach for in vivo microscopy?
It is an important point that is now discussed in the Discussion section (lanes 728-735).
The authors write “It should be noticed that L3 salivary gland nuclei do not contain endogenous Exd given that its nuclear transporter Homothorax (Hth) is repressed by Scr “
It is not entirely clear whether such an effect of Src on the nuclear transport of Exd will be manifested in an ectopic expression system. Could it be this effect that explains the differences in the amount of signal between Scr and Ubx (if Scr suppresses Exd transport, but Ubx does not)?
We showed that VC-Scr and VC-Ubx are expressed at similar levels (new Fig. S1), allowing reaching conclusions on the distinct Scr/Exd and Ubx/Exd BiFC profiles. As mentioned in the main text, the different Scr/Exd and Ubx/Exd profiles are most likely due to different DNA-binding behaviors on the polytene chromosomes (lanes 207-211 and 218-22).
Figure 1 shows the result, which even after deconvolution looks quite fuzzy. Many questions arise about the proof of the specificity of this signal. It would be useful to show what the chromosomes look like when each construct is overexpressed separately.
We are not sure to understand the point raised by the reviewer. Chromosomes look blurry in the enlargement because of the numerical zooming. In any case we did not notice any difference in polytene chromosome organization under the different experimental conditions. We present a new Fig. S1 complementary to the Fig. 1 to better illustrate the similar organization of polytene chromosomes in nuclei expressing or not the VC-Hox fusion protein.
It is noteworthy that in Figure 3 there is a much clearer BiFOR signal for Ubx than in Figure 1
We thank the reviewer for having noticed this subtle difference. We have replaced the BiFOR signal for Ubx in the new Fig. 1.
It would be useful to show the DAPI staining in Figure 3.
LIGHTNING acquisitions with ParB1-mCherry were voluntary performed without DAPI to decrease the acquisition time (already long: 45 min on average for one acquisition) and get stronger BiFC/immunostaining signals (with less rapid bleaching when using two colors instead of three).
In all figures, it would be helpful to add labels (such as arrows) to indicate where the signal is located on all color channels. The colors used when superimposing do not always make it possible to understand how the signals relate in space. For example, when superimposing green on blue in Figure 2, it is not clear where there is colocalization and where there is not. Arrows would help understand the correspondence of signals even without overlap.
We thank the reviewer for this suggestion. We have now indicated more precisely each signal with colored arrows in all figures.
The authors write “ In order to get a better resolution of signals, we applied an adaptative and automated deconvolution method on confocal acquisitions (see materials and methods).” The materials and methods do not contain the word deconvolution; it is not entirely clear that what is described in the materials and methods corresponds to the adaptive and automated deconvolution method
We rephrased the title and added more precision on the LIGHTNING deconvolution (in particular with the fact that it requires special acquisition parameters) in the corresponding paragraph of the Materials and Methods section (lanes 123-141).
Reviewer 3 Report
Comments and Suggestions for Authors
This work by Vanderperre and Merabet demonstrated the use of BiFC in a developing organ (Drosophila larval salivary gland) to visualize and quantify protein-protein interactions at a specific genomic site by combining with the bacterial ANCHOR system. This work took the advantages of the facts that fly salivary gland cells are huge and contain multi-replicated polytene chromosomes. All experiments were well designed with proper negative and positive controls. Statistical analysis appears to be sufficient. However, two concerns need to be addressed:
1). For Figure 3C, results/images shown in C' were generated through immunostaning, not a direct BiFC. Therefore, statistical quantification shown in C'' cannot be labelled as "..BiFC fkh250/..BiFC Total". A revision is needed.
2). On Page 9 and some other places, reference [13] is cited several times to describe a previously published work on how Scr/Exd proteins regulates the activity of an enhancer, fkh250, of the fkh gene in vitro and in vivo. However, currently the reference [13] in References is about Notch signaling and Su(H)/NICD nuclear dynamics. Therefore, a correct reference needs to be provided.
Author Response
This work by Vanderperre and Merabet demonstrated the use of BiFC in a developing organ (Drosophila larval salivary gland) to visualize and quantify protein-protein interactions at a specific genomic site by combining with the bacterial ANCHOR system. This work took the advantages of the facts that fly salivary gland cells are huge and contain multi-replicated polytene chromosomes. All experiments were well designed with proper negative and positive controls. Statistical analysis appears to be sufficient. However, two concerns need to be addressed:
1). For Figure 3C, results/images shown in C' were generated through immunostaning, not a direct BiFC. Therefore, statistical quantification shown in C'' cannot be labelled as "..BiFC fkh250/..BiFC Total". A revision is needed.
We changed the label of the panel (anti-GFP) and of all graphs (to refer to fluorescent intensity) in the new Fig.3 and 4.
2). On Page 9 and some other places, reference [13] is cited several times to describe a previously published work on how Scr/Exd proteins regulates the activity of an enhancer, fkh250, of the fkh gene in vitro and in vivo. However, currently the reference [13] in References is about Notch signaling and Su(H)/NICD nuclear dynamics. Therefore, a correct reference needs to be provided.
We also noticed this mistake and changed the reference 13 with the correct number (ref 20) in the revised manuscript.
Round 2
Reviewer 1 Report
Comments and Suggestions for Authors
This reviewer commends the authors for their extensive revisions to the manuscript. Most of my previous comments have been addressed. My only current recommendation is that the quantifications presented in A"-C" should be presented on one graph and statistically compared to one another. Given the images presented and conclusions reached, there should be statistical differences between localized and non-localized BiFC at the reporter loci.
Author Response
The statistical comparison between the different conditions was shown in the previous Fig. S5. We have now included all statistical comparisons in the new Fig. 3 and 4 (and deleted the corresponding Fig. S5).
Reviewer 2 Report
Comments and Suggestions for Authors
The authors took into account almost all of my comments. I only have a few comments left.
I do not agree with the statement that sgs3-GAL4 driver is less strong than other salivary gland drivers.
Indeed, overexpression of various genes under the control of this driver rarely leads to disturbances in salivary gland development. But this is only due to the fact that the time of expression of this driver is the middle of the third larval instar. The level of expression is extremely high, since the sgs genes are a group of genes encoding the secretion of the salivary gland, and in the middle of the third larval instar these genes give the highest level of expression among all salivary gland genes. I have a lot of experience with this driver. Many of the chromatin proteins we have worked with, when expressed with this driver, have orders of magnitude more binding sites than normal. The reference to Maybeck and Röper 2009 (25) at this point is not very appropriate, since this paper never mentions the sgs driver, so it is not very clear what the authors are referring to when talking about weak expression. Please correct this part of the discussion.
“We showed that VC-Scr and VC-Ubx are expressed at similar levels (new Fig. S1), allowing reaching conclusions on the distinct Scr/Exd and Ubx/Exd BiFC profiles. As mentioned in the main text, the different Scr/Exd and Ubx/Exd profiles are most likely due to different DNA-binding behaviors on the polytene chromosomes (lanes 207-211 and 218-22).”
My question was related to the authors' claim that ScR represses the nuclear transporter for Exd. If ScR suppresses Exd transport but does not suppress Ubx transport, then at the same level of expression one would expect that there would be more Ubx protein in the nucleus, and this could produce more signals. If this is not so, then the phrase about the transporter (“It should be noticed that L3 salivary gland nuclei do not contain endogenous Exd given that its nuclear transporter Homothorax (Hth) is repressed by Scr “) is not clear to me.
Please add the arrows to Figure 2 as to the other figures.
Author Response
Comments and Suggestions for Authors
The authors took into account almost all of my comments. I only have a few comments left.
I do not agree with the statement that sgs3-GAL4 driver is less strong than other salivary gland drivers.
Indeed, overexpression of various genes under the control of this driver rarely leads to disturbances in salivary gland development. But this is only due to the fact that the time of expression of this driver is the middle of the third larval instar. The level of expression is extremely high, since the sgs genes are a group of genes encoding the secretion of the salivary gland, and in the middle of the third larval instar these genes give the highest level of expression among all salivary gland genes. I have a lot of experience with this driver. Many of the chromatin proteins we have worked with, when expressed with this driver, have orders of magnitude more binding sites than normal. The reference to Maybeck and Röper 2009 (25) at this point is not very appropriate, since this paper never mentions the sgs driver, so it is not very clear what the authors are referring to when talking about weak expression. Please correct this part of the discussion.
We amended the corresponding part in the discussion. We deleted the sentence related to the comparative expression level with fkh-Gal4 (we kept the reference 25 that relates to the fkh-Gal4 expression pattern): “Here, we used the Sgs3-Gal4 driver, which is specifically expressed at the mid third instar transition in salivary glands [24]. This driver is more specific than other salivary gland drivers that are expressed at earlier developmental stages and in other tissues (like fkh-Gal4 [25])”.
We also mentioned that Sgs3-Gal3 is strongly expressed, but still allows visualizing distinct binding patterns in salivary gland nuclei: “In addition, despite the strong expression of Sgs3-Gal4, the distinct binding patterns on polytene chromosomes with VC-Scr and VC-Ubx, either alone or in combination with VN-Exd, underlines that our expression system was not at saturation level in salivary gland nuclei”.
“We showed that VC-Scr and VC-Ubx are expressed at similar levels (new Fig. S1), allowing reaching conclusions on the distinct Scr/Exd and Ubx/Exd BiFC profiles. As mentioned in the main text, the different Scr/Exd and Ubx/Exd profiles are most likely due to different DNA-binding behaviors on the polytene chromosomes (lanes 207-211 and 218-22).”
My question was related to the authors' claim that ScR represses the nuclear transporter for Exd. If ScR suppresses Exd transport but does not suppress Ubx transport, then at the same level of expression one would expect that there would be more Ubx protein in the nucleus, and this could produce more signals. If this is not so, then the phrase about the transporter (“It should be noticed that L3 salivary gland nuclei do not contain endogenous Exd given that its nuclear transporter Homothorax (Hth) is repressed by Scr “) is not clear to me.
In his/her first concern, the reviewer was postulating that Ubx could not suppress Exd nuclear transport. We think there is a confusion in mentioning Ubx transport. Here, we explain that there is no endogenous nuclear Exd in salivary gland nuclei, which is independent of the experimental condition.
We have changed our sentence to try to make it clearer: “It should be noticed that L3 salivary gland nuclei do not contain nuclear Exd given that Homothorax (Hth), which is required for the nuclear translocation of Exd, is transcriptionally repressed by Scr at early embryonic stages (by stage 11, [13]; see also https://flybase.org/reports/FBgn0001235 for RNA-seq data in L3 salivary glands). Thus, this observation suggests that Scr could bind less strongly on polytene chromosomes than Ubx in the Exd-free environment of L3 salivary gland nuclei”.
There is no reason to postulate that Ubx could induce nuclear Exd translocation (indirectly by inducing de novo the expression of Hth). On the contrary, there is a common ability of Hox proteins to repress Homothorax, as previously noticed in the antenna imaginal disc: ectopic expression of Antp induces antenna-to-leg transformation due to the loss of Homothorax (and nuclear Exd), which acts as an antenna-selector gene in this tissue (Casares and Mann, Nature 1998, doi: 10.1038/33706). Similar homeotic transformations are observed upon ectopic expression of other Hox proteins, including Ubx (Larsen and Glickman, Dev Genet. 1996; doi: 10.1002/(SICI)1520-6408); Hudry et al., eLife 2014, doi: 10.7554/eLife.01939).
Please add the arrows to Figure 2 as to the other figures.
We added arrows as to the other figures.